# A Framework to Understand Attitudes towards Immigration through Twitter

Yerka Freire-Vidal [1,*], Eduardo Graells-Garrido [1,2] and Francisco Rowe [3]

1   Social Complexity Research Center, Universidad del Desarrollo, Las Condes, Santiago 7610658, Chile; egraells@udd.cl
2   Barcelona Supercomputing Center (BSC), 08034 Barcelona, Spain
3   Geographic Data Science Lab, Department of Geography and Planning, University of Liverpool, Liverpool L69 7ZT, UK; f.rowe-gonzalez@liverpool.uk
*   Correspondence: yfreirev@udd.cl

**Abstract:** Understanding public opinion towards immigrants is key to prevent acts of violence, discrimination and abuse. Traditional data sources, such as surveys, provide rich insights into the formation of such attitudes; yet, they are costly and offer limited temporal granularity, providing only a partial understanding of the dynamics of attitudes towards immigrants. Leveraging Twitter data and natural language processing, we propose a framework to measure attitudes towards immigration in online discussions. Grounded in theories of social psychology, the proposed framework enables the classification of users' into profile stances of positive and negative attitudes towards immigrants and characterisation of these profiles quantitatively summarising users' content and temporal stance trends. We use a Twitter sample composed of 36 K users and 160 K tweets discussing the topic in 2017, when the immigrant population in the country recorded an increase by a factor of four from 2010. We found that the negative attitude group of users is smaller than the positive group, and that both attitudes have different distributions of the volume of content. Both types of attitudes show fluctuations over time that seem to be influenced by news events related to immigration. Accounts with negative attitudes use arguments of labour competition and stricter regulation of immigration. In contrast, accounts with positive attitudes reflect arguments in support of immigrants' human and civil rights. The framework and its application can inform policy makers about how people feel about immigration, with possible implications for policy communication and the design of interventions to improve negative attitudes.

**Keywords:** social network analysis; attitude classification; psycholinguistic analysis; public policy; migration

## 1. Introduction

International migration has emerged as a major divisive global political and societal issue during the 21st Century, with increasing expressions of anti-migration sentiment [1]. Immigration has been portrayed as a major threat to social cohesion, notably during the UK Brexit Referendum and Trump presidential campaign, and drawn attention towards more restrictive migration policies, particularly in Western European countries and the United States [2–4]. Immigration sentiment is also an essential component for successful migrant integration into receiving societies. Discrimination, intolerance and xenophobia can hinder immigrants' capacity to secure employment, housing and achieve a sense of belonging in local communities, contributing to less cohesive societies [5–7]. Global initiatives have been established through the United Nations' Sustainable Development Goals (goal 10) [8] and Global Compact for Safe, Orderly and Regular Migration (goals 16 and 17) [9] to tackle anti-immigrant behaviour and thus facilitate migration integration.

The anti-migration sentiment is generally shaped by misconception [1], and social media has become a key channel to spread misinformation, contributing to the formation of

misconceptions and manifestation of discriminatory acts against immigrants [10]. However, evidence from experimental study designs have revealed that attitudes can be shifted towards a more supportive view of immigration by explicitly addressing misconceptions [11]. Timely access and understanding of public opinion towards migration is thus critical for tackling misconceptions and understanding shifts in local openness to immigrants [12].

Empirical studies on attitudes towards immigration typically draw on survey questions about existing levels of immigration [10]. While surveys are useful, they are an expensive resource in terms of financial cost, labour and time. Moreover, considerable latency may impact data releases, impairing our ability to regularly monitor changes in migration sentiment, identify and tackle prejudice comments against immigrants. However, we know that anti-migration sentiment and prejudice comments can surge during economic recessions [13] and pandemics [14].

Digital trace data sources can now be used to complement and address some of the shortcomings of traditional survey data. Social media platforms, for example microblogging sites such as Twitter, offer a major source of real-time information to understand and quantify attitudes towards immigration. Twitter not only serves as a public forum to exchange opinions and ideas on a broad set of societal issues, including political events [15], abortion legislation [16,17] and migration [18–22], it also shapes the opinions of its users [23]. However, Twitter has also enabled the spread of misinformation and negative rhetoric fueling hate speech [24–26]. Such content has the potential to cause harm to individuals. It often translates into social tension outside the digital world and has played a role in the spread of hate speech during the COVID19 pandemic [14]. With this context in mind, we aim to answer the following research questions: (RQ1) Can we identify, quantify and classify attitudes towards immigration from social network data? (RQ2) What characteristics differentiate the content emitted by users with different attitudes? (RQ3) What emotions and psycholinguistic categories differentiate attitudes?

Using Twitter data and machine learning techniques, we aim to develop a replicable analytical framework to measure and analyse attitudes towards immigration. Specifically, we propose a framework to: (1) identify users' profile stances of positive and negative attitudes towards immigration; (2) analyse the content and psycholinguistic compositions of these profiles; and, (3) monitor their publication activity rhythm over time. We draw on a sample of 160 K tweets and 36 K users discussing immigration in Chile during 2017 when the immigrant population in the country was recorded to have increased by a factor of seven since 2002 from 105 k to 746 k, with over half of new arrivals occurring between 2012 and 2017 [27].

Our contributions are three-fold: first, we propose a methodological framework to operationalise mainstream theories of social psychology on the formation of attitudes towards immigration using Twitter data. It enables identifying users' positive and negative stances on immigration based on their public opinions and characterising the content and psycholinguistic features of these stances. Second, our proposed methodology reveals how digital traces can be used to complement and augment traditional data sources by enabling understanding of short-term changes in attitudes towards immigration and multidimensional views of immigration sentiment. Finally, our case study provides valuable insights into our limited knowledge of the patterns, experiences and challenges of recently arrived immigrants in Chile. Specifically, our work offers insights into the formation of attitudes towards immigration in Chile during a period of large migration influx, largely related to human displacement in Colombia and an exodus from Venezuela.

The paper is structured as follows. Section 2 discusses the theoretical and conceptual background related to our research work. Section 3 describes the dataset used for analysis. Section 4 describes the proposed methodology before Section 5 presents the results. Section 6 discusses the implications and limitations of our work, and Section 7 offers some concluding remarks.

## 2. Background

In this section, we review theoretical approaches to the formation of attitudes towards immigration. Then, we describe the literature on measuring attitudes based on Twitter data to illustrate the key challenges and advantages of using this data source. Finally, we describe immigration in Chile.

### 2.1. Theories and Measurement of Attitude Formation

Two theoretical models are often used to describe the formation of attitudes towards immigration. These are the Intergroup Contact Theory (ICT) [28] and the Integrated Threat Theory (ITT) [29,30]. ICT explains how *positive* attitudes form, while ITT describes how *negative* attitudes are created.

The ICT postulates that increased social interaction among people from different population subgroups reduces prejudice and enhances trust. The theory is based on conditions such as common goals for different groups and the benefits of cooperation and support (both formal and informal) to reach those goals. Existing research has contributed evidence to support these arguments [31–33]. Increased intergroup interaction reduces fear and anxiety that may exist when people interact with individuals from an unfamiliar group [34,35] by promoting empathy and understanding [36,37].

In contrast, the ITT predicts that social interaction among people from different groups may lead to perceptions of threat and contempt toward members of the different groups [29,30]. Two types of threat describe the formation of negative attitudes: symbolic and realistic. In the context of migration, these threats are related to competition in the labour market, to public health concerns from possible diseases, to increased crime and physical well-being, and to perceptions of the size of the foreign group, among others [38,39], as well as an increased fiscal burden [40].

The common method to measure attitudes toward immigration is to use public opinion survey data. Some examples include the Gallup World Poll, the Pew Global Attitudes Survey, the International Social Survey Programme, the World Values Survey, the Iposos Global Trends, the European Social Survey, and the Eurobarometer [41]. Data from the Gallup World Poll revealed that in all major regions of the world, people are more likely to want immigration to remain at the current level or increase, rather than decrease, with the exception of Europe. However, there is great variability between countries. In Europe, for example, southern Europeans tend to show more negative attitudes towards immigration than northern Europeans, who show more positive attitudes [4].

Although there is now a large repository of public opinion surveys on migration, some important limitations with these traditional methods arise. For example, existing surveys are mainly from European and North American countries, the number of questions collecting opinions on migration is highly variable across countries, their frequency is coarse, and they are often costly to implement. Thus, it has become a necessity to obtain frequent, comprehensive information covering different regions of the world to understand attitudes toward immigration, as stated in the first objective of the Global Compact for Safe, Orderly and Regular Migration: "to collect and use accurate, disaggregated data to formulate evidence-based policies" [9].

In this work, we propose that the discussion in social networks offers a novel and inexpensive source to complement traditional data systems. It can be collected in near real-time, which would allow for richer analyses grounded on the aforementioned theories. In the following, we describe the potential of the social network Twitter in this regard.

### 2.2. Social Media Analysis in the Study of Human Behaviour

To discuss social media analysis from three perspectives: human behaviour, attitude classification and psycholinguistic analysis. We focus on the analysis of the micro-blogging platform Twitter.

Researchers from different disciplines have used data extracted from the Twitter platform to study psychological and socio-cultural characteristics of communities. Some ex-

amples include: personality differences between Democrats and Republicans [42], influence of diurnal and seasonal variability on mood [43], happiness associated with Christianity/atheism [44], political polarisation [45], influence of culture on personal actions [46], prediction of attitudes in response to a trigger event [47], measurement of the level of integration of immigrants in different cities of the world [48], among others. When the analysis is about groups of people, Twitter data highlights its potential use in the detection of hate speech [49], i.e., threatening, harassing or seriously offensive language, as well as to characterise hateful users [50]. The detection of hate speech against immigrants is a particular case of the general framework of hate speech detection. Several studies have used Twitter data to study hate speech against immigrants [18–22]. C. Arcila C. et al. [22] model and characterise anti-immigrant hate speech on Twitter in Spain. They find that hate speech against immigrants includes Islamophobia, rejection of public support towards the immigrant group, and has a greater presence of offensive than violent language.

In this context, we develop a mixed approach grounded on the theories about attitude formation instead of following a hate speech approach, which is limited to hate but not necessarily opposition/approval or feelings of threat/empathy toward migration. Each formation theory defines an attitude, and, in cases where the classifier confidence is low, we define an *undisclosed* stance to account for participation in the debate without disclosing attitude [51]. Particularly, we build upon our previous work to classify users into attitudes as political stances using a tree-based classifier [17]. As stances, attitudes are not always explicit, and thus, they must be predicted. Two types of features are commonly used for prediction. On the one hand, stance can be predicted using network interactions, based on the assumption that like-minded individuals are more likely to interact [52,53]. On the other hand, lexical analyses have shown to allow predicting stance as vocabularies within stances tend to have strongly associated words [54,55], and even other non-textual cues such as emojis [17].

A related area to attitude classification is sentiment analysis. In general, such methods extract the polarity of a text in terms of positive, neutral, and negative intents [56,57]. These measures may be correlated to attitudes, however, empathetic tweets about migration could speak negatively about discrimination, and threat-related tweets could speak positively about the country, making the interpretation of the results difficult. However, we perform one type of emotional analysis when characterising the content published by users in each attitude. We use the Linguistic Inquiry and Word Count (LIWC) psycholinguistic lexicon [58] to distinguish multidimensional aspects of the migratory discussion. These aspects include multiple emotions and other topical dimensions of expression, which may be compared between attitudes. LIWC usage with social media data includes identifying health issues [59,60], predicting political sentiment [61], and understanding perception about the transportation experience [62]. The Spanish version has been validated experimentally [63], enabling its consideration in our work.

### 2.3. Immigration in Chile

In the Latin American context, Chile presented the greatest increase in the weight of the immigrant population between the last two censuses [64]. Although immigration is not a new phenomenon in Chile, it has increased dramatically in recent years and has become a relevant issue in the national debate (see Figure 1). The immigrant population grew from 1.27% of the total population in 2002 to 4.35% according to the 2017 Census, with a more diverse origin for all immigrants than in previous years. As of 2017, 66.7% of all resident immigrants arrived in Chile from 2010, and 61% arrived in 2015–2017 (considering up to 19 April 2017, the date of application of the census).

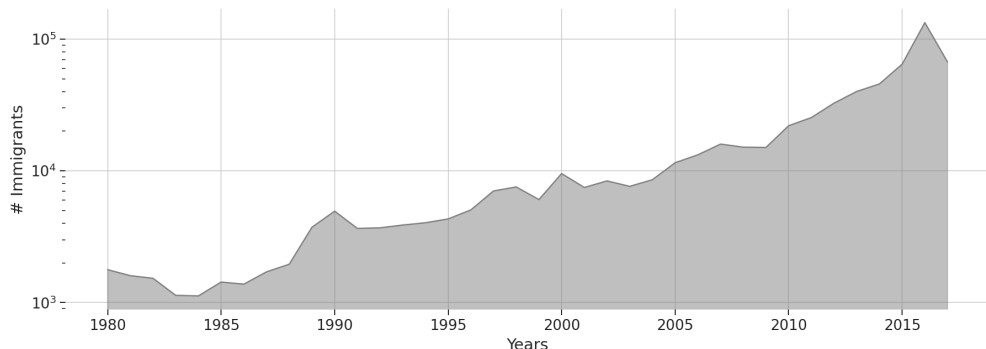

**Figure 1.** Number of immigrants in Chile by year of arrival according to the 2017 Census.

There were two milestones that impacted the national agenda and installed migration as a social concern [65]. First, in August 2016, the Chilean Public Ministry opened an investigation into the possible involvement of an airline in a case of migrant trafficking. Second, in July 2017, the Ministry of Health confirmed a case of leprosy in a Haitian citizen. Leprosy is a disease without records in the country except for overseas islands [66].

A national-level survey in Chile revealed that 57% of nationals agree that the country should take more drastic measures to exclude illegal immigrants, versus 19% who disagree. In addition, 41% of respondents agree that immigrants raise crime rates, compared to 38% who disagree with this statement. In the area of labour competition, 40% of respondents agree that immigrants take jobs away from Chilean-born people, and 36% disagree. This survey was applied in the months of April-May 2017 [67]. Arguably, the questions are more related to the ITT than the ICT. To the extent of our knowledge, there are no other surveys that try to understand how Chileans feel about migration. Hence, this scenario puts Chile as a relevant case study with respect to measuring local attitudes towards immigration. Such analyses may provide complementary knowledge of the attitudes and perceptions of the population, first, by including social theories, and second, by providing fine-grained, dynamic insights.

## 3. Dataset

In this section, we describe the Twitter platform and the dataset used to measure attitudes towards immigration.

Twitter is a micro-blogging platform where users report a screen name, a full name (which can be real or fictitious), a location (real, fictitious, or empty), and a small autobiography, among other attributes. Each user publishes posts (called *tweets*) with a maximum of 280 characters. Twitter also allows interaction between its users: users can *follow* others users and to see their tweets in their own timelines. Users can mention other users in their own tweets using a handle (e.g., @username), *quote* other tweets or adding commentary to them, or *retweet* (RT) another tweet to share it with one's audience.

To crawl tweets that discuss immigration in Chile, we connected to the Twitter Streaming API using a system designed to crawl Chilean tweets [68]. The query parameters were keywords related to immigration, e.g., "inmigración" (*immigration*), "inmigrante" (*immigrant*), "fronteras" (*borders*), "racismo" (*racism*), etc.; and origin countries with their respective denomyms. Thus, the dataset after data cleaning is composed by 160,775 tweets (54,252 are plain tweets and 106,532 are retweets—RTs–. In addition, 20,248 tweets are quotes and 19,265 are replies) that are on topic during 2017, written by 36,698 users (see Figure 2 for the temporal distribution of content).

The cleaning process ensured that the discussion under analysis was about human migration in Chile. Examples of noise topics were: racism toward indigenous groups in Chile, bird migration, a South American soccer championship, national presidential elections, migration issues in México, the U.S.A., and Spain, among others. Hence, we excluded tweets from users with a reported (or predicted) location different to Chile, as well as tweets in languages other than Spanish.

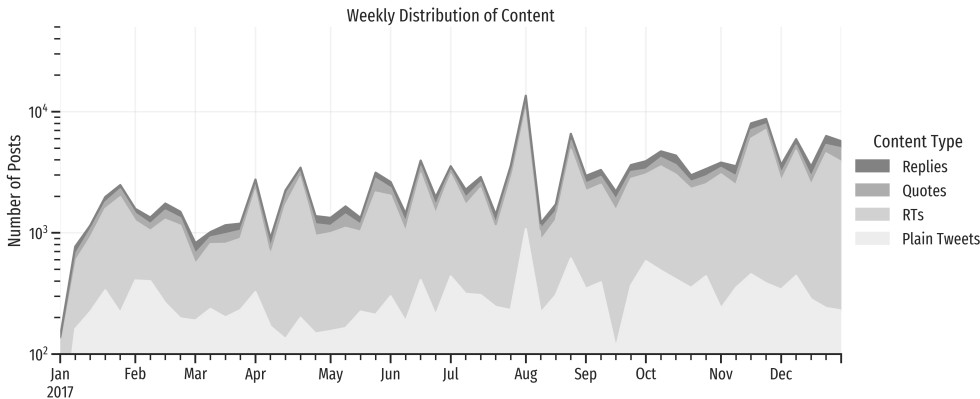

**Figure 2.** Weekly volume of content in the dataset, disaggregated by type of content/interaction.

## 4. Methodology

In this section, we describe how to classify, quantify and characterise the attitudes toward immigration. The methodology is composed of the following steps: theory-informed profile tagging, which enables to pinpoint some users with an attitude; propagation of user attitudes to the rest of the dataset, after training a classifier with the tagged users; and then perform a characterisation of attitudes from the lens of users, content, psycholinguistics, and dynamics. A schematic diagram of the methodology is shown in Figure 3.

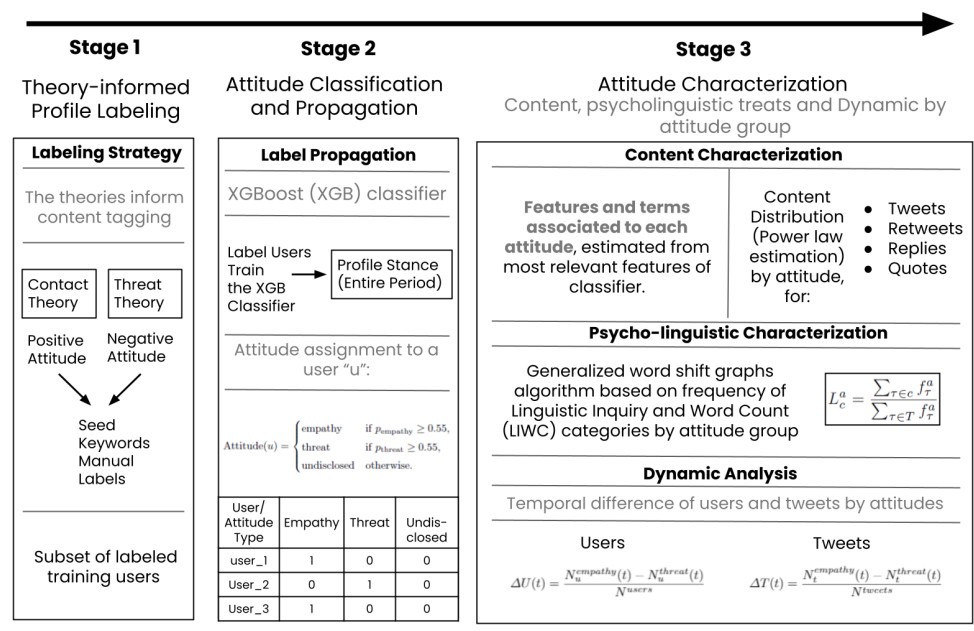

**Figure 3.** Schematic diagram of the proposed methodology.

### 4.1. Theory-Informed Profile Labelling

Here we define the two attitudes we classify profiles into, and how to label profiles according to these attitudes for classification. In the spectrum of attitudes towards immigration there are two that define two opposite extremes. Usually, these types of categories are named *positive/negative*, *in-favour/against*, or similar. Here we rely on the social psychology theories described in Section 2: the Intergroup Contact Theory (ICT), and the Integrated Threat Theory (ITT). Based on those theories, we name the two attitudes we are classifying as *empathy*, due to the empathy toward immigrants, and *threat*, due to perceptions of threats regarding migration.

Given the potential size of the discussion under analysis, manually labelling the user profiles (or their tweets) into these two categories is expensive and impractical. In view

of this difficulty, we predict attitudes using a classifier trained on a labelled subset of the dataset. This subset is labelled automatically from a list of seed patterns and keywords for each attitude, as words are an effective mechanism to predict the community a user belongs to [18,69,70]. To identify seeds, we iteratively explore the dataset to seek for features that could be mapped to the *empathy* and *threat* attitudes. In *empathy*, we look for features that indicate that immigrants are welcome and will be received in equal conditions (e.g., "we are all immigrants"). In *threat*, we look for features that show that immigrants are not welcome or qualify them or their arrival negatively (e.g., "illegal immigrants take our jobs"). The labelled terms are not necessarily frequent, instead, they are discriminative, i.e., it is likely that someone in its corresponding category would use the term, and not from the other. The list is built iteratively in the sense of running the first steps from this section up to the classification step, and then exploring the usage of discriminative terms by people in each group to look for other potential seeds. See example seeds for each attitude in Table 1.

**Table 1.** Seed patterns for each attitude toward immigration.

| Attitude | Seed Words and Hashtags (and Their Frequencies in Dataset) |
| --- | --- |
| Empathy | #bienvenidosachile (*welcome to Chile*, 26), #chilesinbarreras (*Chile without barriers*, 7), #chilediverso (*Diverse Chile*, 43), #noalaxenofobia (*no to xenophobia*, 126), #nomasxenofobia (*no more xenophobia*, 30), #racismo (*racism*, 1336), #stopracismo (*stop racism*, 48), #nomasracismo(*no more racism*, 10), #noalracismo (*no to racism* 73), #pongamonosinmigrantes (*let us become immigrants*, 67), #todossomoshermanos (*we are all siblings*, 157), #todossomosmigrantes (*we are all migrants*, 1108), #bienvenidosmigrantes (*welcome migrants*, 6), #ningunserhumanoesilegal (*no human being is illegal*, 16), #nadieesilegal (*no one is illegal*, 29), #nohayserhumanoilegal (*no human being is illegal*, 22), #redmigrante (*migrant network*, 105), #interculturalidad (*interculturality*, 218), #díadelmigrante (*migrant's day*, 269), #sinfronteras (*without borders*, 371), inhumano (*inhuman*, 349), multicultural (372), diversidad (*diversity*, 1531) |
| Threat | #inmigrantesilegales (*illegal immigrants*, 7), #nomasinmigrantes (*no more immigrants*, 41), #vendepatria (*sells homeland*, 5), #estadodecatastrofe (*state of catastrophe*, 44), invasión (*invasion*, 1258), invaden (*they invade*, 62), turba (*group of people generating chaos*, 732), prestamistas (*moneylenders*, 372), narcotráfico (*drug trafficking*, 531), turistas (*tourists*, 2186), fronterizo (*at the border*, 292), enfermedades (*diseases*, 2138), narcotraficantes (*drug dealers*, 291), expulsarlos (*eject them*, 105), echarlos (*to take them out*, 1172), deportarlos (*deport them*, 88) |

Next, we label users who match these patterns. Those who are labelled in both attitudes have their labels removed. Note that we assume that in the period under study attitudes do not change. Additionally, we manually label accounts of institutions (such as the International Migration Office and the Jesuit Service for Migrants), opinion leaders, journalists, and politicians that have explicitly expressed their attitude on the issue.

### 4.2. Attitude Classification and Propagation

To predict attitudes, we follow a bootstrapped approach, where we propagate the user labels from the previous step to the rest of the dataset. We use the XGBoost classifier that trains decision trees using gradient boosting [71]. The input feature matrix is the concatenation of several matrices:

- A content-term matrix, where each row represents user $i$, and each term $j$ can represent a word, hashtag, username, URL or emoji. Thus, a cell $(i, j)$ contains the number of times user $i$ has used the term $j$ in their tweets.
- A profile-term matrix, analogous to the previous one, but this time for the terms contained in the full name and biographical self-description of each user.
- A profile-domain matrix, mapping to each user's home page its main domain (e.g., twitter.com) and their main top level domain (e.g., .com).
- Since homophily may vary or be absent in different interaction layers [72], we consider the three types of interaction separately. Thus, we build three adjacency matrices based on the interactions in the discussion: retweets, replies, and quotes. Each matrix stores in a cell $(i, j)$, the number of times user $i$ has interacted with user $j$ (for instance, if $i$ retweets $j$ one time, $c(i, j) = 1$).

- A user–attitude interaction matrix for each type of interaction, where each cell contains the number of times the corresponding user has interacted with other users that were labelled with an attitude.

Then, we train the classifier using the set of labelled users. To avoid overfitting, we take two measures. First, the gradient boosting is performed with early stopping, using a validation set of 15% of the training observations. Second, we removed columns from the feature matrix that were used for labelling. This includes the seed keywords for each attitude, as they perfectly separate users from both groups and our goal is to classify users who do not use these terms in their content.

After having trained the classifier, we predict the attitude of the rest of the dataset. For a given profile $u$, the classifier outputs a value $p_a(u)$ for each attitude $a$ that lies in $[0, 1]$, corresponding to the fraction of decision trees that vote for the corresponding attitude. Note that the value of $p_a(u)$ is not a real probability. Thus, we apply a small threshold to consider predictions with a number of voters higher than a random choice. Those users who cannot be classified are marked as *undisclosed*. As a result of this stage, we assign an attitude to a user $u$ according to the following function:

$$\text{Attitude}(u) = \begin{cases} \text{empathy} & \text{if } p_{\text{empathy}} \geq 0.55, \\ \text{threat} & \text{if } p_{\text{threat}} \geq 0.55, \\ \text{undisclosed} & \text{otherwise.} \end{cases}$$

After predicting attitudes, we manually check for profiles that are highly active/followed in the discussion and could have been mislabelled by the classifier. We add those manual labels and then repeat this stage.

### 4.3. Attitude Characterisation

In the last stage, we characterise each attitude from multiple perspectives. We describe what characterises each attitude from the lens of content (what is published in each category and how?), the lens of psycholinguistics (which semantic categories discriminate the expression of emotions in each category?), and the lens of dynamics (when does each attitude express their opinions?).

#### 4.3.1. Content Characterisation

To measure what is published in each category and how, we focus on two aspects of content: the profile features that are most associated with each attitude, and the distribution of tweet vocabulary per attitude. On the one hand, the association of the features deemed important for classification provides insight into which ones are associated with each attitude [17]. However, the classifier estimates plain feature relevance, without any association to each attitude. Hence, for all relevant features, we estimate their association to each attitude using the log-odds ratio with uninformative Dirichlet prior [73], a measure that weights features in a similar way to TF-IDF, with the addition of controlling the variability of frequency. We apply this weighting to three documents, one per attitude (including *undisclosed*). Each document is the column-oriented sum of the feature matrix for all users predicted in the corresponding attitude. Here, we expect to find different terms that express the same concept but are associated with opposite attitudes [70], as well as differences in features from the self-reported biographies, and even in emoji usage [17].

On the other hand, we know that the content generation in Twitter tends to follow a powerlaw [74,75], that is, their distributions can be described approximately as $P(x) \sim x^{-\alpha}$. Hence, for each attitude, we characterise the volume of content (the number of plain tweets, RTs, quotes, and replies) according to the exponent $\alpha$ of their fitted powerlaw distributions. These exponents enable us to compare if attitudes behave differently in their discussion and interaction mechanisms.

### 4.3.2. Psycholinguistic Characterisation

The previous lens provided differences in the content published by each attitude. However, a direct content-based approach does not discriminate the expression of emotions in each category. To gain a deeper understanding of attitudes in this direction, we use a well-known psycholinguistic lexicon named "Linguistic Inquiry and Word Count" (LIWC) [76]. LIWC was designed to capture emotional, cognitive, and structural components present in text. It is available in several languages. Its Spanish version contains 7515 words classified in one or more of 72 categories that belong in four dimensions:

1. Standard linguistic processes: articles, prepositions, pronouns, etc.
2. Psychological and affective processes: positive and negative emotions, with subcategories such as anger and anxiety.
3. Relativity: time, verb tense, motion, space.
4. Personal matters: sex, death, home, occupation, etc.

We will focus our analysis on LIWC categories that are possibly associated with some of the factors shaping empathetic and threatening attitudes towards immigration. With regard to *empathy* attitudes, we consider the following LIWC categories as relevant: affective processes, positive feeling and emotions, optimism and energy, humans, social, family, and inclusion. With regard to *threat* attitudes, we consider the following LIWC categories as relevant: anger, anxiety, negative emotions, inhibition, death, body, job, and money. We estimate the association between attitudes and LIWC categories. First, we define the association of LIWC category $c$ to attitude $a$, $L_c^a$, as the relative frequency of words in $c$ with respect to all terms in the discussion per attitude:

$$L_c^a = \frac{\sum_{\tau \in c} f_\tau^a}{\sum_{\tau \in T} f_\tau^a},$$

where $T$ denotes the vocabulary, and $f_\tau^a$ the total frequency of vocabulary term $\tau$ by accounts with attitude $a$. To explore these associations, we visualise them using Generalised Word Shift Graphs [77]. These visualisations summarise each attitude according to the differences in association between attitudes and LIWC categories, with the aim of describing the emotional and semantic aspects of attitude formation.

### 4.3.3. Dynamics and Events

Our final lens of characterisation aims to understand when each attitude expresses its opinions. Although we assume that attitudes are constant in the time under study, there still could be differences regarding how they are expressed in time in terms of volume. Particularly, we compute the daily difference between the number of tweets and the number of users per attitude. Positive values of this difference indicate a tendency towards *empathy*, whereas negative values indicate a tendency towards *threat*. Then, we compare these time-series with a null-model where we randomly permute the predicted attitudes for users 1 K times. If for a given day, the time-series does not intersect the 95% confidence interval of the null series, we consider it different with statistical significance. Finally, by identifying dates with salient and significant differences, we can explore what triggers the expression of each attitude. In this work, we manually do this through visual exploration.

The proposed methodology provides a full pipeline to characterise users attitudes towards immigration in a micro-blogging platform. Next, we apply this methodology to a set of tweets in Chile during 2017.

## 5. Results

This section first presents the results associated with the classification and characterisation of user attitudes towards immigration according to *empathetic* attitudes and *threatening* attitudes. Next, we analyse the content composition of these user profiles before examining differences in their lexical expressions and temporal fluctuations in attitude sentiment.

### 5.1. Attitude Identification and Classification

To answer our question RQ1, we performed an exhaustive search in the dataset for hashtags and words that could be strongly associated with each attitude (see Table 1 for more examples). For instance, the hashtags #bienvenidosachile (welcome to Chile) and #inmigrantesilegales (illegal immigrants) can be directly mapped to each attitude. We found more hashtags in the *empathy* group than in the *threat* group. When tagging threat users, we included words that are not normally associated with threats or negative comments against immigrants. For instance, "turistas" (*tourists*) is not a word commonly associated with a threat attitude. However, in our dataset, tourists tended to be used to express anger about people entering the country on a tourist visa and overstaying.

In total, 3.1 K accounts were tagged in the empathy group, and 1.2 K accounts were tagged in the threat group (see Table 2). The attitude classifier presented good performance, with high precision for both attitudes (0.95 in *empathy*, 0.81 in *threat*) and high recall (0.88 in both). Such good performance is expected, due to a tagging strategy based on a perfect separation of accounts from each attitude. However, this strategy works well identifying a tendency towards attitudes because the keywords used for tagging are excluded from the learning process [17].

**Table 2.** Performance (precision and recall) of the user profile classifier based on 10-fold cross-validation.

| Attitude | Precision (mean) | Precision (std.) | Recall (mean) | Recall (std.) | Labelled Accounts |
|----------|------------------|------------------|---------------|---------------|-------------------|
| Empathy  | 0.95             | 0.04             | 0.88          | 0.13          | 3118              |
| Threat   | 0.81             | 0.16             | 0.88          | 0.12          | 1233              |

Figure 4 presents the top-50 features according to their importance in classification. These features include: retweets to accounts that were be automatically tagged with each attitude, mentions to then-president Michelle Bachelet (@mbachelet), attitude-relevant terms such as cesantía (*unemployment*), delincuentes (*criminals*), inmigración (*immigration*), migrantes (*migrants*), and haitianos (*Haitians*); and mentions to migrant-support institutions as @sjmchile (Jesuit Center for Migration). The remaining features in the top-50 include interactions with authors of viral tweets, and terms relating to specific content, such as a quote by Umberto Eco.

We then used the classifier to propagate attitudes to the rest of the dataset (88% of the total dataset). Table 3 reports the resulting classification. Considering the labelling and propagation processes, a total of 71.98% user profiles were classified in *empathy*, 3.24% in *undisclosed*, and 24.78% in *threat*. This is an unexpected outcome. We expected the distribution of user profiles to be negatively biased given the social divisive nature of immigration in Chile. However, this may imply that negative users are more vocal than positive users and hence there is a perception for negative comments to dominate the public discussion around immigration issues in Chile. Table 3 reveals this is the case with the threat group generating considerably more content per user (with a ratio of 3.5 tweets per user, for the empathy attitude group, versus 6.7 tweets per user, for the threat attitude group) than the empathy group, particularly quotes and replies.

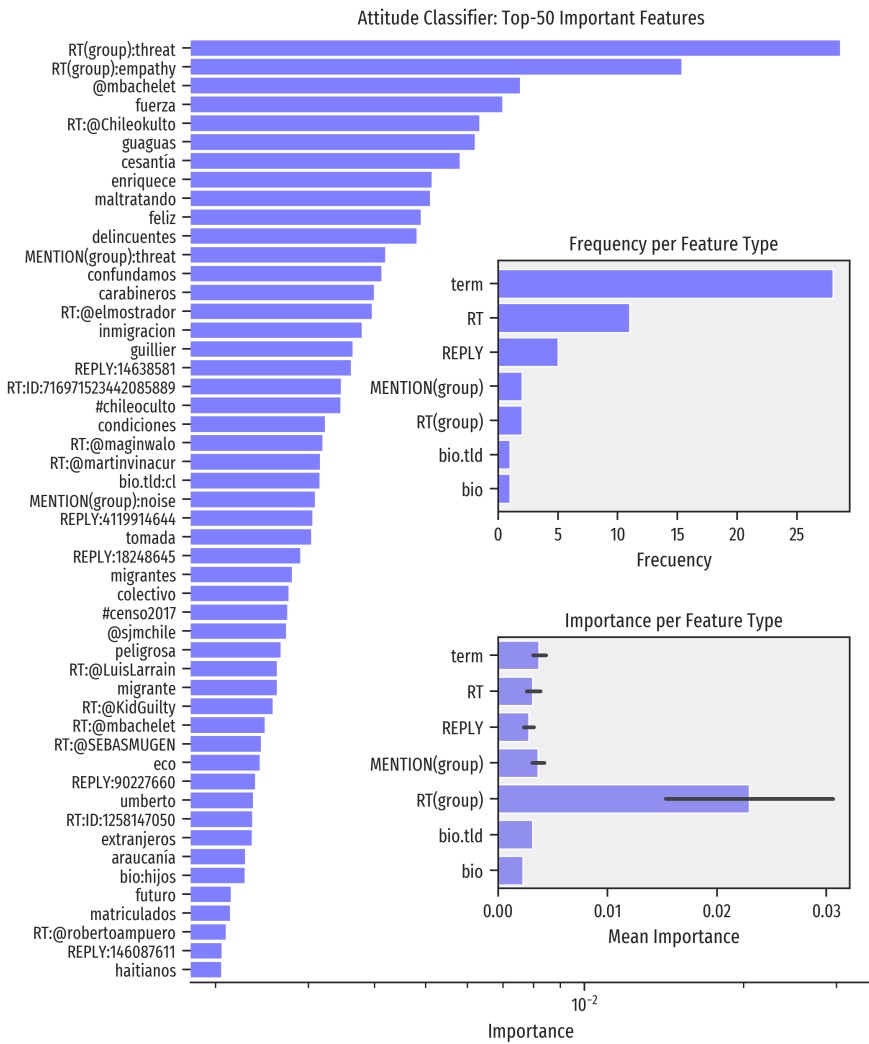

**Figure 4.** Attitude classification: Top 50 most important features.

**Table 3.** Distribution of user accounts and tweets per attitude and account type.

| Attitude | Accounts | Total Tweets | RTs | Quotes | Replies |
|---|---|---|---|---|---|
| Empathy | % 71.98 (26,414) | % 57.46 (92,374) | % 61.94 (65,989) | % 45.62 (9238) | % 32.81 (6320) |
| Undisclosed | % 3.24 (1190) | % 4.52 (7261) | % 1.44 (1531) | % 1.87 (379) | % 3.61 (695) |
| Threat | % 24.78 (9094) | % 38.02 (61,140) | % 36.62 (39,012) | % 52.50 (10,631) | % 63.59 (12,250) |

### 5.2. Content Characterisation

To better understand the discussion driving the expression of positive and negative attitudes towards immigration and answer our question RQ2, we performed content analysis producing a characterisation of the most frequently used terms associated with each user stance profile. We weighted each term using log-odds ratios with an Uninformative Dirichlet Prior [73]. Figure 5 displays the top-30 attributes per attitude, revealing that own-attitude retweeting behaviour is the most prominent feature for *empathy* and *threat*. This is arguably expected due to homophilic interactions [72], a phenomenon that has been observed in political discussion in Chile [78]. However, care must be taken when interpreting the result, as this reflects interactions with accounts that were pre-labelled only. A plausible explanation is that highly popular accounts that can be labelled through methods tend to make their positions explicit.

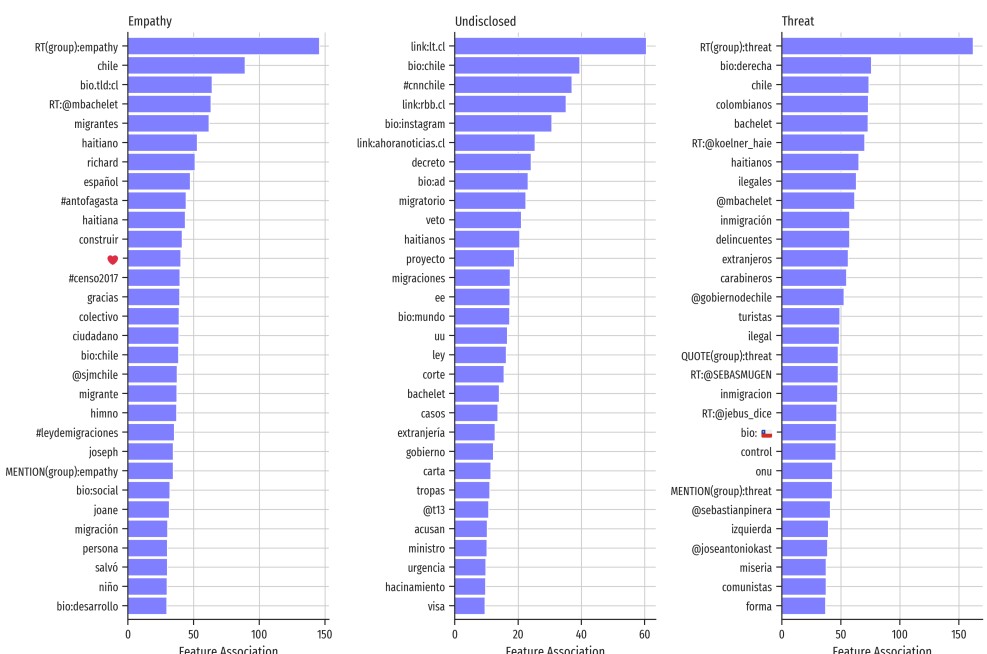

**Figure 5.** Top-term associations to each attitude (or lack thereof), estimated using log-odds ratio with uninformative Dirichlet prior.

Neutral terms, such as migrante (*migrant*) and migración (*migration*) and haitiano (*Haitian* as a singular noun) feature prominently in the *empathy* group. Emojis are typically used to express political attitudes [17] and we observe that the red heart emoji is associated with *empathy*, possibly as an expression of solidarity or liking of immigrants-related content. Content related to news items also feature prominently in empathetic Twitter content. Ranking in the top-50 items, we identified the case of Richard Joseph: a Haitian who saved a person that fell from a nine-story building; and, Joane Florvil: a Haitian mother who died after being arrested by the Chilean police under unclear circumstances. We also identified political-based debates, including retweets of Michelle Bachelet's and mentions of a debate of a national migration law (#leydemigraciones).

In contrast, words such as inmigración (*immigration*), extranjeros (*foreigners*) and haitianos (plural, *Haitians*) are often used by the threat group. Immigration placed the focus on migration as a process and concerns about its potential implications for the national health, education and labour market systems, rather than on understanding the individuals themselves. The use of foreigners and Haitians may be used to draw a clear distinction between the "we" and "us". The use of the Chilean flag also features prominently among the *threat* group, arguably associated with expressions of nationalism. *Threat* users also tend to identify their alignment with right-wing views, including "derecha" (right-wing) in their biographies; tagging right-wing politicians, such as Sebastián Piñera (@sebastianpinera, current president of Chile) and José Antonio Kast (@joseantoniokast, extreme right-wing presidential candidate); and, posts about left-wing parties ("izquierda") and the Communist Party ("comunistas"). Overall, the results highlight a strong association between anti-migration views and conservative political ideologies. This is despite not including any explicit political keywords in the seed list for our stance classifier.

We did not observe incidence of the seed words (used to train the classifier) in the content analysis and relevant features associated with each attitude; since only one of them appears within the top ranking (tourists, in the threat group). Note that the total number of training seed words is small (56 in total; 34 for the empathy attitude, and 22 for threat) and their frequency of use is mostly small (see Table 1 for some examples).

To identify and quantify differences in the diffusion of content generation, by both groups of attitudes, we analysed the respective distributions of the number of tweets, retweets, quotes and replies. Because other studies suggest that these measures of content

generation follow a powerlaw distribution [75,79], we fit powerlaw regressions on the distributions of the aforementioned measures. Figure 6 reports the results. Theoretically, a symmetrically diagonal line would indicate perfectly equal spread generation of content. A perfectly horizontal line would indicate high concentration of the content generation by a single user. Figure 6 reveals a relatively symmetrical distribution in the generation of content for empathetic users across tweets, retweets, quotes and replies. This contrasts with the distributions associated with the threat group that displays higher concentration of content generation in a small number of users, with consistently lower powerlaw exponents to empathetic users. This is particularly prominent for retweets suggesting a high degree of interaction within the social network of threat users. These findings confirm our previous interpretations that while our dataset includes a small number of threat users, they are more vocal and a minority of these users tend to generate a comparatively larger amount of content than empathy users.

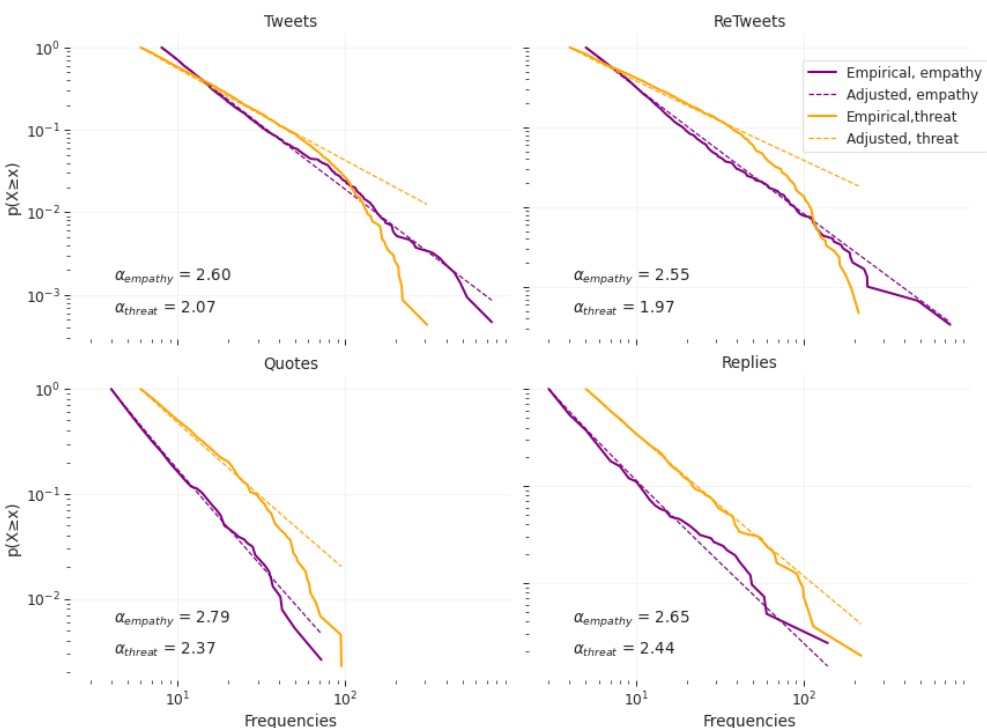

**Figure 6.** Powerlaw distributions for content metrics.

### 5.3. Psycholinguistic Characterisation

We also analysed the manifestation of cognitive and emotional structures in the text based on the LIWC lexicon and answered our question *RQ3*. Figure 7 displays the differences in the relative frequency of the LIWC categories between each group, i.e., *empathy* and *threat*.

The results reveal that categories linked to social, posemo (positive emotions), school, comm (communication), humans, optim (optimism and energy), family, affect (affective processes), incl (inclusion), and posfeel (positive feelings) are found more often in the *empathy* group. These are all psycholinguistic concepts used to describe empathetic attitudes associated with contact theory hypotheses. Conversely, categories related to motion (defined by words such as move, walk, go out), othref (reference to other people), present, negemo (negative emotions), death, inhib (inhibition), anger, money, anx (anxiety), and job are more commonly found in expressions used by the *threat* group. This is consistent with the threat theory, as immigrants can be perceived as an economic threat and labour competition.

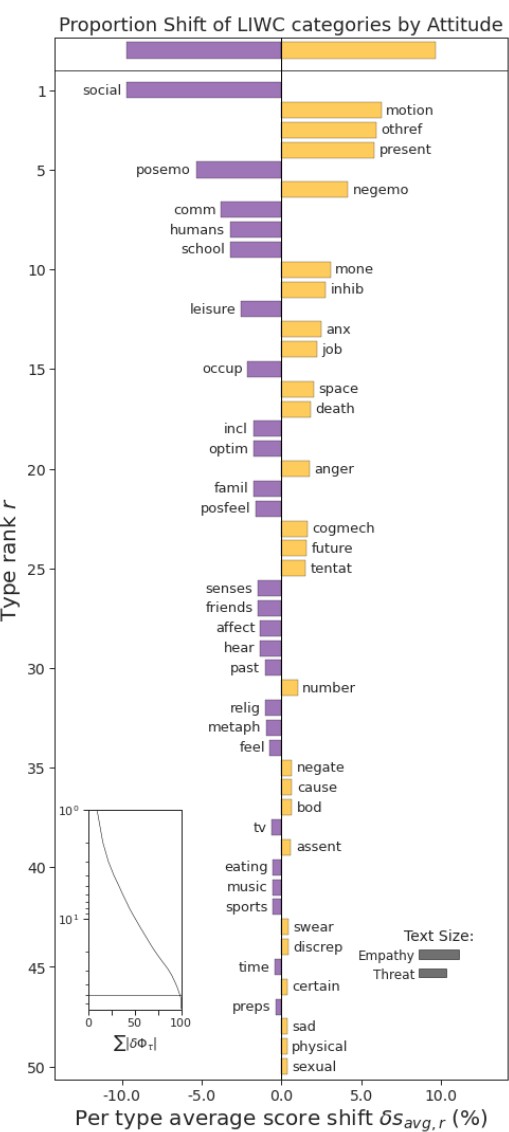

**Figure 7.** Proportion shift of LIWC categories in tweets grouped by *empathy* (on the left in purple) and *threat* (on the right in orange) attitude.

### 5.4. Dynamics and Events

Finally, we examined changes in user engagement with the migration debate over time. We estimated the daily difference in participation rate between user profile stance; that is, proportion of users in each stance group actively engaging with Twitter content about migration (see Figure 8). A positive difference indicates greater engagement among *empathy* users than *threat* users, with negative values denoting greater engagement among the latter. The resulting differences are small, though notable discrepancies exist on specific dates, particularly in the second half of 2017. Peak discrepancies in engagement coincide with key migration-related events that received mass news media coverage:

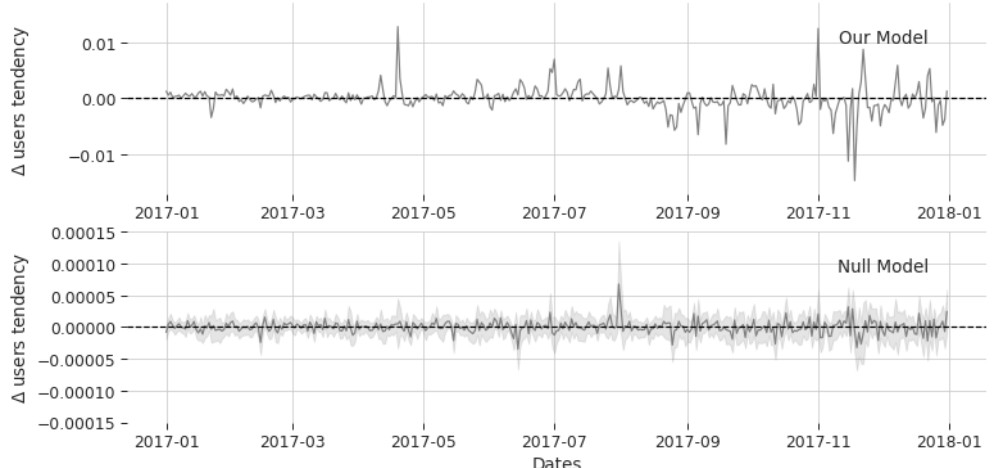

**Figure 8.** Daily difference in the number of users between *empathy* and *threat* groups. Top: our model, and bottom: the null model.

- **19/04:** The Chilean Census was carried out. Tweets regarding foreign interviewers went viral (positive trend spike).
- **01/07:** The National Institute of Human Rights filed a complaint against a migrant trafficking gang (positive trend spike).
- **06/09:** The Minister of Home Affairs, Mario Fernández, faced an appeal at the National Congress after a delay in the Migration Law (negative trend spike).
- **19/09:** A sign was installed in Talca (a city on central Chile) urging Haitians to join the Communist Party (negative trend spike).
- **01/11:** Director of the Central Public Hospital declared that Joane Florvil (Haitian woman who died after being arrested by Chilean police) had been beaten at the police station (positive trend spike).
- **15/11:** Senator Fulvio Rossi from Antofagasta (a city in northern Chile) was stabbed. He stated that "the attacker had a foreign accent and would be a black person" (negative trend spike).
- **18/11:** Haitian immigrants attacked police in a commercial neighbourhood in downtown Santiago (negative trend spike).
- **22/11:** (1) The Court declared the posthumous innocence of Joane Florvil. (2) Michelle Bachelet recognised not only the heroic act of Richard Joseph (Haitian citizen who rescued a woman who fell from a building) but also a set of positive human values in migration (positive trend spike).

We performed a similar analysis estimating the daily differences in the number of tweets between threat and empathy groups (see Figure 9). The results from this analysis display similar patterns of spikes as those found examining differences in user engagement. However, it highlights two key events broadcasted by the national news media: (i) Michelle Bachelet's announcement of a new visa for migrant children and youth through the "Chile welcomes you" program on 26 July (positive trend spike); and (ii) a confirmed case of an Haitian citizen with leprosy in Valdivia (a city in southern Chile) on July, 31st (negative trend spike).

We validated the differences in both measurements by comparing them with a null model, where attitudes were assigned at random (maintaining the original distribution of attitudes) in 1 K dataset permutations. Figures 8 and 9 display the null model timeseries. Given that the identified peaks are far from the 95% interval of each null model, the patterns described here are significant.

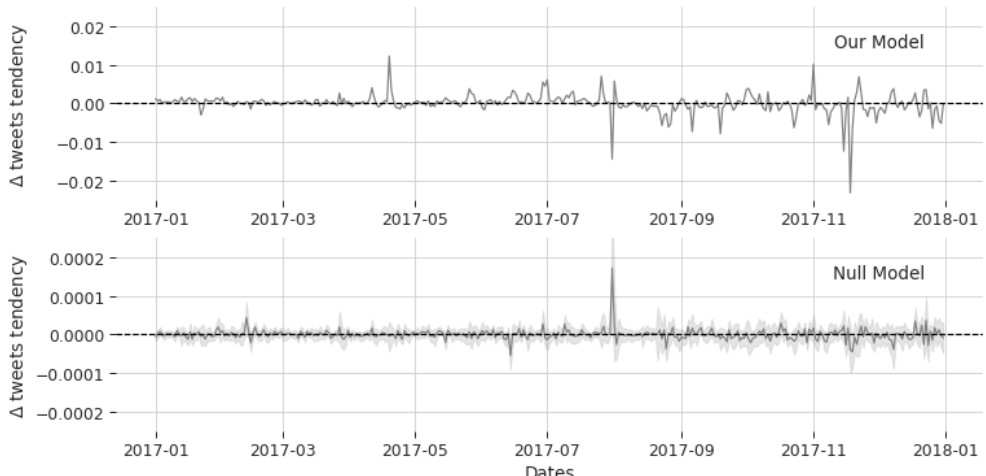

**Figure 9.** Daily difference in the number of tweets between *empathy* and *threat* groups. **Top**; our model, and **bottom**; the null model.

These results highlight the important role of national news media outlets in shaping the formation and expression of attitudes towards immigration in Chile. All the key events identified above were broadcasted via national television and featured on major national newspapers. As highlighted in intergroup contact and threat theories, the way in which these events are portrayed may contribute to the formation and intensified manifestation of empathetic and threatening attitudes on social media. For example, highlighted cases of physical attacks by immigrants may foment threat perceptions to public safety and hence invite expressions of negative behaviour towards immigration. The case of the Haitian citizen with leprosy is an interesting example, coinciding with intensification of tweet content generation by *threat* users and exceeding that produced by *empathy* users, despite a greater number of the latter group engaged in the discussion.

In contrast, other events portrayed by the news media were met with empathetic reactions. These include: the creation of a new visa for migrant children and young people, the case of a migrant trafficking gang, of a Haitian citizen who died while being arrested by the police, and immigrants being depicted as victims of violence and discrimination. A prominent example is also the case of a Haitian citizen who rescued a Chilean woman featuring how immigrants can contribute to the local society.

## 6. Discussion

### 6.1. Key Results

Social media is a new, dynamic and open space to express opinions and feelings, collect data and better understand public perception of immigration and offers the opportunity to overcome key limitations of traditional data sources. Social media offers the potential to monitor online public opinion about migration in near-real time at unprecedented spatial and temporal granularity, and understand the actual extent of empathy or negativity in these opinions in raw format. To unlock these potentials, we developed a novel, reproducible and open framework to measure and monitor changes in immigration sentiment. Particularly, the proposed framework enables the classification of users' stances of positive and negative attitudes towards immigrants and characterisation of these profiles quantitatively summarising users' content and differences in engagement and content generation at a daily temporal scale.

We presented evidence of the composition of the Chilean immigration sentiment network on Twitter. We found that a larger share (72%) of the user network displays positive and empathetic sentiment towards immigration, and that the proportion of the network associated with negative immigration attitudes was small (25%). However, we also found that users displaying negative anti-immigration profiles tended to produce content at a significantly greater rate, producing up to 50% more content per user than

users displaying positive immigration profiles. We find interesting the apparent contrast between public surveys held in Chile and our findings: reportedly, a majority of Chilean nationals agree with the following statements: "the country should take more drastic measures to exclude illegal immigrants." However, the survey is measuring something different: agreement with a statement regarding a specific group of people rather than the expression of an attitude toward a phenomenon. Furthermore, the population under study is different, as the survey samples the population aiming at national representativeness. Thus, a future line of work should be the measurement of representativeness of the Twitter population, and the definition of a methodology to compare survey results with results from our work.

We present a characterisation of the feature and psycholinguistic content of positive and negative immigration user profiles. Users with a pro-immigration profile often use neutral terms recognising migration as a process, and emotional content relating to social, positive emotions, communication, school, humans and family. By contrast, users with an anti-immigration profile tend to use terms denoting a difference between immigrants (*them*) and the native population (*us*), and emotional content relating to motion, other people, present, negative emotions, death, inhibition, anger, money, anxiety and work. We also found evidence of strong alignment between users with anti-immigration views and conservative political ideologies. Our temporal analysis also revealed pronounced peaks in daily user engagement and content generation activity in response to key migration-related events, particularly events featured in news media.

### 6.2. Interpretation

Our findings of a dominant base of users with pro-immigration profiles are consistent with existing prevalent trends in most of the world's regions [80]. Our findings also suggest that although the user base with anti-immigration profiles may be small, it produces and disseminates content at a significantly faster rate. Similar to the effect generated by fake news, the degree of novelty of anti-immigration content and resulting emotional reactions may be the cause of its rapid spread and generation [81].

We also showed that empathetic user profiles were linked to positive emotions, inspiring respect and unity largely in support of immigrants' human and civil rights. Anti-immigration user profiles were associated with negative emotions, calling for stricter immigration laws and claims of migrants "stealing" jobs from locals. We also found evidence of strong alignment between anti-immigration user profiles and conservative political ideologies. This pattern is not specific to Chile, but it is prevalent across industrialised countries [4].

Our analysis revealed high variability in daily user engagement and content generation activity as a result of key migration-related events. These events were prominently featured in national news media, highlighting the pivotal role that news media outlets may play in shaping the formation and expression of attitudes towards immigration in Chile. This calls for a careful approach in the way in which news media outlets portray news items involving immigrants.

### 6.3. Limitations and Future Work

There are two key aspects that need further exploration and that limit the scope of our results: the dynamic analysis of LIWC categories associated with attitudes, and the representativeness of Twitter. In terms of dynamics, it would be interesting to study the temporal distribution of psycholinguistic categories within empathic and threatened attitudes. This would enable quantifying the potential influence of news events on attitudes. This would provide a way to measure the effect of events and their depictions and narratives on how people feel with respect to migration.

In terms of representativeness, we acknowledge that Twitter is a biased sample of the population [82]. However, Twitter is one of the most widely used applications in Chile [83], and it reflects some of its cultural aspects, such as the centralisation of the

country [84]. Furthermore, a Twitter-based analysis of the abortion discussion in Chile was found to present equal insights as those from the main national survey that covered the issue [85], hinting that there are social insights that can be derived from the platform. Thus, we propose that this work provides insights with respect to the discussion, although the representativeness of such insights is yet to be determined.

One line of work we sought to explore was to conduct a spatial analysis of attitudes, to understand the relationship between attitudes and the actual presence of immigrants in a place. This would provide a way to measure real and imagined threat attitudes [86], link virtual and physical places of expression and coexistence in a single analytical framework, and allow us to identify socio-demographic characteristics associated with the various inferred attitudes. However, as documented in the literature [10], the spatial representation of Twitter data is limited. Less than three per cent of tweets are geolocated [87]. We recognise that addressing these biases is an active area of research, and case-specific weighting schemes have been proposed to ensure the statistical and spatial representativeness of social media data [88]. However, developing weighting schemes requires knowledge of social media users' profiles. While this information can be obtained with some level of accuracy from Facebook, access to Twitter users' personal attributes is very limited. Therefore, we cannot guarantee that our results represent the general population.

## 7. Conclusions

In this paper, we present an analytical framework for monitoring attitudes towards immigration. Specifically the proposed framework enables the classification of users' stances of positive and negative attitudes towards immigrants and characterisation of these profiles quantitatively summarising users' content and temporal stance trends.

We applied the proposed framework to 2017 Twitter data from Chile, to capture changes in the virtual public discussion about migration during a period of a surge in immigration. We presented evidence of positive *empathetic* attitudes being expressed by a broad group of users, representing expressions of support for the immigrant community. Particularly these supportive expressions relate to calls for respect, dignity and treatment of immigrants' human and civil rights. Conversely, we provided evidence revealing that negative *threatening* attitudes towards immigration emerge from a reduced number of users, and that these attitudes are prevalent in discussions calling for stricter migrant regulation and concerns about labour competition. We also showed that negative attitudes are more commonly manifested and tend to intensify during instances of negative portrayals of immigrants. These results suggest that media news outlets play a critical role in the spread of negative representations of immigrants, and highlight the need for a more careful approach in the way in which events involving migrants are communicated. Media news outlets should consider the potential impact of misinformation fuelling misconceptions and prejudiced behaviour against immigrants. More broadly, our results demonstrate the need for a systematic approach to monitor immigration sentiment and identify shifts in attitudes towards immigrants. Such approach can enable rapid and effective mitigation plans to address misconceptions and prejudice comments against immigrants and to cushion the long-term formation of negative migration attitudes and their detrimental impacts on national social cohesion.

**Author Contributions:** Conceptualisation, Y.F.-V., E.G.-G. and F.R.; methodology, Y.F.-V. and E.G.-G.; software, E.G.-G.; validation, Y.F.-V., E.G.-G. and F.R.; formal analysis, Y.F.-V.; investigation, Y.F.-V., E.G.-G. and F.R.; resources, E.G.-G.; data curation, E.G.-G.; writing—original draft preparation, Y.F.-V.; writing—review and editing, Y.F.-V., E.G.-G. and F.R.; visualisation, Y.F.-V. and E.G.-G. All authors have read and agreed to the published version of the manuscript.

**Funding:** This research received no external funding.

**Institutional Review Board Statement:** Not applicable.

**Informed Consent Statement:** Not applicable.

**Data Availability Statement:** The tweet identifiers and the classification results needed to replicate the results from this paper are available in the following shared folder: https://drive.google.com/drive/folders/1MmZyNMVtxpgcCyNhLy6UgCZjFALYi72r?usp=sharing, accessed on 16 September 2021.

**Conflicts of Interest:** The authors declare no conflict of interest.

## Abbreviations

The following abbreviations are used in this manuscript:

| | |
|---|---|
| API | application programming interface |
| ICT | intergroup contact theory |
| ITT | integrated threat theory |
| LIWC | linguistic inquiry and word count |
| RQ | research question |
| RTs | retweets |
| TF-IDF | term frequency—inverse document frequency |
| UK | United Kingdom |

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
