# Peer review of "A Framework to Understand Attitudes towards Immigration through Twitter"

_applsci, doi:10.3390/app11209689_

Round 1

Reviewer 1 Report

Overall a very interesting and relevant contribution. With a very good presentation of the limits of the methodologies usually used to study attitudes towards immigration, a very good justification of the analysed field nd a very well presented methodology to analyze these attitudes towards immigration via Twitter in Chile. However, three points deserve to be better explained (even if they are addressed). The question of the representativeness of the content on Twitter : the discrepancy between the results drawn from the study and those that can be drawn from survey studies is from this point of view to be discussed. The question of the articulation between the methods of identifying attitudes in matters of immigration and the study of the vocabulary used by the authors of tweets: is it not this vocabulary which partly contributes to the identification of attitudes? ? The question of changes over time in attitudes to immigration. The methodology makes it possible to analyze the state of expression of opinion in immigration matters on Twitter as a function of time. Does this say anything about changing attitudes towards immigration? It seems questionable and should be better emphasized within the limits of the work. Which does not alter its interest. 

Reviewer 2 Report

Many thanks for giving me the opportunity to read this paper. I particularly enjoyed the novelty and felt that the paper is a good piece of work using a clear framework to detect tweets on the topic of immigration in Chile. The authors also use some placebo tests to relate the impact of media on the nature of tweets. I felt that this part was super interesting and that the paper does not give enough focus on the relationship between traditional media and social networks like Twitter, while it was highly interesting. 

Introduction

The introduction is clear. Contributions and results are clearly highlighted, and past work are reviewed. I am however wondering whether the author could be more precise about the use of Twitter in general: is it used to shape other followers' opinions or just to express one's beliefs ? In the former, Twitter is equivalent to social media while in the latter, it just reveals the opinion of its users, and it is equivalent to a survey. Porcher and Renault (2021) discuss this point in a different framework and you might refer to their work, and test this proposition.

Theory

As you talk about groups / coalitions in theory, I am wondering whether we can geotagg tweets and map users' sentiments relative to immigration depending on the color (blue or red) of the municipalities in which they live. More precisely, are conservative regions more against immigration? Or more sensitive to media announcements on immigration ? 

Methodology

The methodology section is sound to me. Some of the limitations of the sample / methods are discussed in the end of the paper.

Results

Results are presented using different word associations and dynamics in time. I believe that you should organize the theoretical arguments in order to answer these different research questions. It would give more powerful insights to your results.

Extra references:

Porcher and Renault (2021), PLOS One. https://doi.org/10.1371/journal.pone.0246949

Reviewer 3 Report

The article proposes a framework to analyze sentiments towards immigration. Overall, the article is well written and organized. The presentation of both the framework and the result analysis is in great detail. 

Author Response

We thank the reviewer for his/her appreciation of the paper.

Round 2

Reviewer 2 Report

The authors responded to all my comments.